# *"This is not my body"*: Therapeutic experiences and post-treatment health of people with rifampicin-resistant tuberculosis

**Marian Loveday[1,2]***, **Sindisiwe Hlangu[1], Lee-Megan Larkan[3], Helen Cox[4],**
**Johnny Daniels[5], Erika Mohr-Holland[5], Jennifer Furin[6]**

**1** HIV Prevention Research Unit, South African Medical Research Council, KwaZulu-Natal, South Africa,
**2** CAPRISA-MRC HIV-TB Pathogenesis and Treatment Research Unit, Doris Duke Medical Research
Institute, University of KwaZulu-Natal, Durban, South Africa, **3** Greytown Specialised TB Hospital, KwaZulu-
Natal Department of Health, Pietermaritzburg, South Africa, **4** Institute for Infectious Disease and Molecular
Medicine and Wellcome Centre for Infectious Disease Research in Africa, University of Cape Town, Cape
Town, South Africa, **5** Médecins Sans Frontières, Khayelitsha, Cape Town, South Africa, **6** Department of
Global Health and Social Medicine, Harvard Medical School, Boston, MA, United States of America

* marian.loveday@mrc.ac.za

doi.org/10.1371/journal.pone.0251482

University, CHINA

**Data Availability Statement:** These are qualitative
interviews and they do contain information that
could be identifying. Thus we cannot publish them
or make the widely available since this could lead

## Abstract

### Background

There are few data on the on post-treatment experiences of people who have been success-
fully treated for rifampicin-resistant (RR-)TB.

### Objective

To describe the experiences and impact of RR-TB disease and therapy on post-treatment
life of individuals who were successfully treated.

### Methods

In this qualitative study in-depth interviews were conducted among a purposively selected
sample from a population of individuals who were successfully treated for RR-TB between
January 2008 and December 2018. Interview transcripts and notes were analysed using a
thematic network analysis which included grounded theory and a framework for understand-
ing pathophysiological mechanisms for post-TB morbidity and mortality. The analysis was
iterative and the coding system developed focused on disease, treatment and post-treat-
ment experiences of individuals. This paper follows the COREQ guidelines.

### Results

For all 12 participants interviewed, the development of RR-TB disease, its diagnosis and the
subsequent treatment were a major disruption to their lives as well as a transformative expe-
rience. On diagnosis of RR-TB disease, participants entered a liminal period in which their
lives were marked with uncertainty and dominated by physical and mental suffering. Irre-
spective of how long ago they had completed their treatment, they all remembered with clar-
ity the signs and symptoms of the disease and the arduous treatment journey. Post-

to inadvertent recognition of the participants, since the rich quotes and descriptions cannot be fully anyonymized. Our ethics committee is willing to share the data with individuals who request it. Contact can be made by reaching out to Ms Anitha Gupta the research administrator (email: Anitha.gupta@mrc.ac.za) at the ethics board at the South African Medical Research Council.

**Funding:** The authors received no specific funding for this work.

**Competing interests:** The authors have declared that no competing interests exist.

treatment participants reported physical, social, psychological and economic changes as consequences of their RR-TB disease and treatment. Many participants reported a diminished ability to perform physical activities and, once discharged from the RR-TB hospital, inadequate physical rehabilitation. For some, these physical limitations impacted on their social life, and ultimately on their psychological health as well as on their ability to earn money and support their families.

## Conclusion

The experiences and impact of RR-TB disease and therapy on post-treatment life of individuals successfully treated, highlights gaps in the current health care system that need to be addressed to improve the life of individuals post-treatment. A more holistic and long-term view of post-TB health, including the provision of comprehensive medical and social services for post-treatment care of physical ailments, social re-integration and the mitigation of the perceived fear and risk of getting TB again could be a central part of person-centred TB care.

## Background

Tuberculosis (TB) is a leading cause of global morbidity and mortality, with the World Health Organization estimating that 10 million people fell ill with TB in 2019, of whom 1.4 million died [1] TB programmes tend to focus on narrowly-defined bacteriologic indicators to determine if treatment for the disease has been "successful" [2], and there are limited data on ongoing morbidity and disability among people who have survived TB treatment. Studies have demonstrated a higher mortality rate among TB survivors when compared with the general population [3–5] as well as ongoing medical problems, including reduced pulmonary function, reactive airway disease, and an elevated risk for recurrent episodes of TB [6–8]. As currently structured, however, TB programmes make few accommodations for managing these ongoing health issues [9].

In addition to the physical consequences of TB, the disease is also associated with a significant burden of psychological distress [10][0] Some of this is due to the highly stigmatizing nature of TB [11], while additional mental health stresses come from socioeconomic deprivation, loss of usual roles/identities, and challenges coping with the prolonged and difficult treatment [12]. Few, if any, resources are offered to survivors to address the long-term mental health burden associated with surviving TB [13] and little has been documented regarding this aspect of post-TB recovery, a significant oversight in this era of purported "patient-centered care" [14].

The manner in which people are able to cope with the consequences of serious diseases after completion of therapy is significantly influenced by their experience during treatment [15]. The concept of "liminality"—often referred to as a "state of in-between-ness" [16]—has been used as a framework to understand the experiences of survivors of a range of diseases, including cancer [17], chronic renal disease, and HIV [18]. In this framework, sick people have an initial experience that marks a disruption in their lives: as they seek to return to the state of health they experienced prior to the diagnosis of their illness, they must undertake specific tasks (often referred to as "rites of passage" and which usually encompass treatment and its effects). These experiences are often transformative in that the individuals undergo major changes in their physical and psychological status resulting from the illness. The things they

experience in the liminal period leave them forever altered and are important in shaping the long-term implications of their struggle with their disease [19]. Understanding the experiences of people living with TB during this liminal period between diagnosis and "cure" could help identify post-treatment support strategies for both the physical and mental health sequelae of this pervasive infectious disease [20].

Little is known about the post-treatment experiences of people who have survived the more serious and difficult-to-treat form of TB known as rifampicin-resistant tuberculosis (RR-TB). Because there are limited data on this topic, we conducted an exploratory qualitative study nested within a larger cohort study that was investigating mortality and TB recurrence among individuals after reported successful RR-TB treatment. Using a liminality framework, the goal of this sub-study was to describe the experiences and impact of RR-TB disease and therapy on individuals who were successfully treated.

## Methods

### Study design

This was a qualitative study done using open-ended interviews. Individuals successfully treated for RR-TB between January 2008 and December 2018 were identified in the national treatment register. Individuals were purposively selected to ensure both males and females were represented and that there was representation across the study period.

### Study setting and population

The study was conducted in Umzinyathi district in KwaZulu-Natal South Africa. Umzinyathi, a rural district with poor infrastructure, is considered one of the poorest districts in South Africa [21],[1] with high burdens of HIV, TB and RR-TB. From January 2008 to December 2018, 1316 individuals with RR-TB were treated at the Greytown RR-TB hospital, 77% of whom were co-infected with HIV. Of note, all patients initiated on RR-TB in this district were hospitalized to initiate treatment before being discharged to continue treatment in the community. Over this time period the successful treatment rate was 67%.

### Data collection and analysis

An initial 27 individuals were identified, but only 15 could be contacted. Three declined participation in the study, two as they had relocated to another area a considerable distance away and the third having recently started a new job was unable to take a day off. In total, welve people participated in open-ended interviews using a semi-structured guide (See Appendix 1) in a private office setting with no other individuals present. The interview guide was designedto ask them about their experiences of their RR-TB disease, its treatment and their life since completing treatment. All interviews were conducted by two female authors (ML and SH with more than two decades of collective experience in open-ended interviewing) in the language in which the participant felt most confident (isiZulu or English). The interviews were recorded and transcribed into English for analysis. Participants were told about the interviewers' interest in the topic matter as part of the formal consent process and that the interviewers were both employees as full-time researchers. Each participant was interviewed only once with interviews lasing between 30 and 90 minutes. Field notes were not kept or analyzed.

Data analysis was based in grounded theory, which centers the analysis on the accounts of the study participants, as opposed to using an already-existing analytic framework [22]. Grounded theory was selected as little is known about the post-treatment experiences of persons who have been successfully treated for RR-TB. As noted in the introduction, however, the

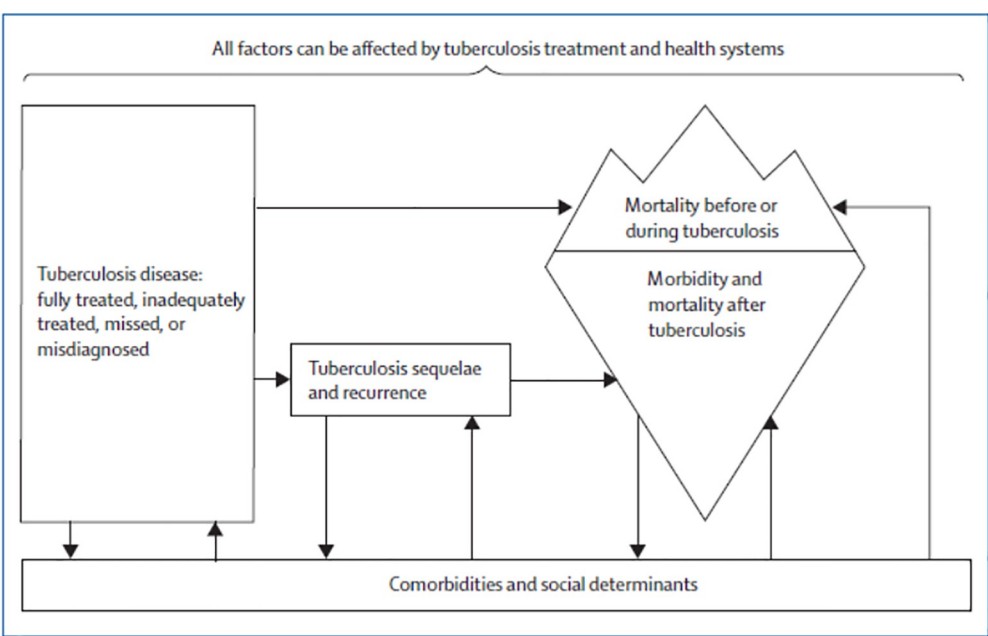

**Fig 1. Biological framework for understanding post-TB treatment morbidity and mortality developed by Datta and Evans [24].**

analysis was also based in the concept of liminality [23]. Further informing our theoretical approach to the data analysis was the framework for understanding pathophysiological mechanisms for post-TB morbidity and mortality proposed by Datta and Evans and summarized in Fig 1 below [24]. This framework, however, was modified and adapted based on the interviews done with the study participants to yield the final analytic framework used in this study, which is shown in Fig 2.

A thematic network analysis was performed on the study interviews and transcripts [25, 26]. After an initial review of the data during which participants described the experience of their illness and how this affected their life during and after treatment, a coding system was developed by one study team member (ML). This analytic framework was verified/modified by another author (JF), and the interviews analyzed. Discrepancies were resolved via discussion and there was agreement among all study team members on the final analytic framework used (see Fig 2). Interviews were halted after 12 participants since inductive thematic saturation had been reached (determined by two team members, JF and ML) [27], as no new codes or themes were emerging in the dataset [28]. Data collection, analysis, and reporting for this qualitative study followed the consolidated criteria for reporting qualitative research (COREQ) guidelines [29]. Of note that while the interviews were focused on the post-treatment experiences of the participants, most participants framed such experiences within the context of what happened to them during treatment. Thus we also present information on treatment experiences as a way to frame the post-treatment experiences of the participants.

## Ethics

Written consent was obtained from all the patients willing to participate in the study. The consent included participation in the interview and digital audio recording, the voluntary terms of involvement in the study and the assurance of confidentiality and anonymity. Patient anonymity was maintained by identifying each patient using a unique identification number. Ethical

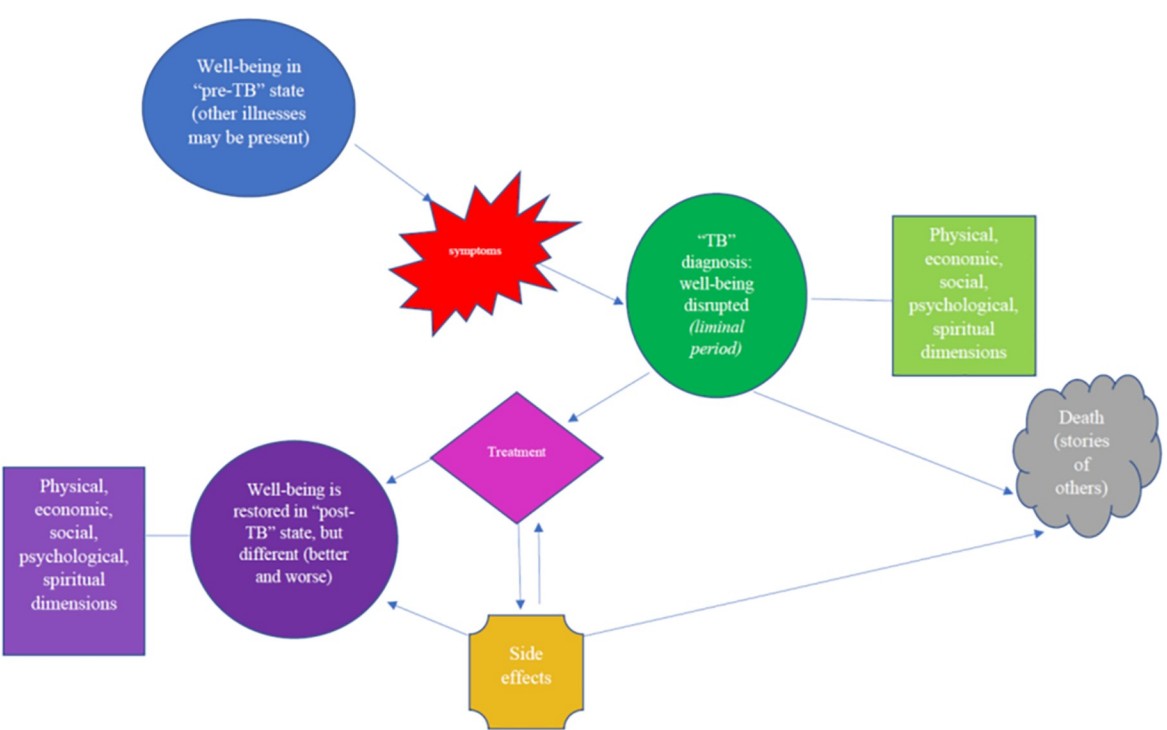

**Fig 2. Enhanced analytic framework developed and utilized in this study.**

approval was obtained from the South African Medical Research Council (SAMRC) Ethics Review Committee (EC010-6/2018) and the KwaZulu-Natal Health Research Committee.

## Results

Twelve participants were interviewed. Their demographic data and information about their health backgrounds are presented in Table 1. All patients were considered "successfully treated."

**Table 1. Details of study participants.**

| Study no | Age | Gender | Resistance pattern | HIV status | Year Rx ended |
|---|---|---|---|---|---|
| 1 | 35 | F | MDR-TB | Positive | 2014 |
| 2 | 70 | F | RR-TB | Negative | 2018 |
| 3 | 47 | F | MDR-TB | Negative | 2015 |
| 4 | 33 | M | MDR-TB | Positive | 2016 |
| 5 | 62 | M | MDR-TB | Negative | 2011 |
| 6 | 52 | M | MDR-TB | Positive | 2017 |
| 7 | 35 | F | MDR-TB | Positive | 2017 |
| 8 | 43 | F | MDR-TB | Positive | 2010 |
| 9 | 30 | F | MDR-TB | Positive | 2013 |
| 10 | 40 | F | Rif mono-TB | Positive | 2015 |
| 11 | 30 | F | RR-TB | Positive | 2014 |
| 12 | 33 | M | MDR-TB | Positive | 2013 |

Abbreviations: Rx, treatment; MDR, multidrug resistant TB; TB, tuberculosis; RR-TB, rifampicin-resistant TB; Rif mono-TB, rifampicin-mono resistant TB.

## Well-being disrupted: TB diagnosis and hospitalization

For all participants interviewed, the development of RR-TB disease, its diagnosis and the subsequent treatment were all major disruptions to their lives. The symptoms that heralded TB, the events that occurred at the time of diagnosis, and the initial treatment experience of hospitalization—which was required for treatment to begin—were frightening and bewildering. The prospect of death all too real.

Irrespective of how long ago the participants had completed their treatment (between one and seven years), they all described with clarity the signs and symptoms of the disease they had experienced as the initial indication that something was wrong. The most common signs and symptoms described prior to diagnosis included loss of weight, cough and shortness of breath.

> *"I got thinner and had longer fingers. I had gone weak and I just wished for nothing in life. I hated dawn. You know when its sunrise, eish! I hated that bird that makes noise at dawn. And I would think, 'Oh no, I'm alive, I'm not dead!!'"*

--52 year-old male

> *"When I initially went there I was losing weight and I was coughing but I didn't understand that cough because I was treating it with many different medicines, but it wasn't going away…..."*

--35 year-old female

These worrisome changes in their physical well-being marked a departure from their previous lives. These physical changes were reinforced by the medical system which imposed other changes to their lives. Chief among these was hospitalization. For all the participants in this study, the initiation of treatment for the RR-TB required physical relocation to the hospital environment. This move was an important "rite of passage" into the new period they were entering in their lives. For some, the forced and prolonged hospitalization was a source of great distress and made it difficult for them to "return to normal". As one participant reported:

> *"As sick as I am, I take [the medications]. When I got here [to the hospital], I was treated well, I began to see the nurses coming to see me for chats. And my mind started to freshen up. They explained to me, there was a class where we learned about this disease. I understood clearly and accepted. I told my family about the disease and that it is curable after how long. But what made me sad was being told that I was going to stay here for 6 months, but it was just shock. I am going to be here for this long?"*

--30 year-old male

And another stated:

> *"Eish! It took a long time, and I think what made me sick the most was that I was not somebody who was used to being in hospital."*

--33 year-old male

## Returning to well-being: The treatment journey

**Physical issues.**    Physical problems and challenges dominated the lives of study participants during their illness experience and throughout the liminal period. While some

participants blamed their TB disease for their health problems, others reported that it was the treatment for RR-TB as well as the hospitalization that caused most of their physical distress.

As one participant noted:

*"When I started MDR-TB pills* [sighs] *it was difficult, in fact it was extremely difficult because my body wasn't responding too well! So, I think that's one of the reasons I was kept here for a long time. . . ehh TB treatment was mistreating me, a lot. So, I had a problem with just looking at them. There is this trolley they come with, it is a medication trolley and as soon as I see it coming I won't be able to eat."*

--35 year-old female

And another stated:

*"I think what made me sick the most was that I was not somebody who was used to being in hospital. . .That's what made me sick, because it took a very long time. Although I had started treatment, but I was still very sick."*

-33 year-old male

The adverse effects of treatment significantly interfered with even the most basic of activities, leaving many of the participants unable to do the day-to-day activities they took for granted prior to becoming sick:

*"I could not go to fetch water because I could not lift the bucket, I could not stand for a long time because my feet were painful. . . I was well when I came here in hospital, I could do all the household tasks. But when I was discharged from hospital, I could not go fetch water and I could not cook because my body was aching."*

--47 year-old female

And as another participant stated:

*"When I urinated and my bed* [was] *wet, I would stay like that, because I couldn't do anything for myself, I couldn't even go to the bathroom to bath myself. I would sit and keep quiet, if nobody came to visit me from home, I would just stay like that."*

--30 year-old female

Participants reported their strong desire to leave this "helpless" state behind and to move into a period of greater independence and functionality. This desire motivated many of them to take their treatment, even though they felt the treatment itself was the cause of many of their problems. Why did the participants continue to feel treatment was necessary in order to leave their transitory "sick selves" behind, even when it caused them significant physical and social discomfort? Some participants reported great faith in the medications, and despite the large pill burden, found the act of "pill taking" to be a part of the passage to health. As one participant stated when asked if she ever tired of taking her treatment:

*"Yes, it sometimes happened but there was that thought of 'what is going to help me' if I stop taking them. Just looking at them, the way they were many, when you have to swallow them. They were 17* [pills] *if I'm not mistaken. Just looking at them in front of you when you have to*

*start swallowing them. It used to come but then you know very well that this is your life, what are you hoping for if you stop taking them? What can I say, [silence for few seconds] they (pills) gave me hope because I felt the difference day by day if I swallow them."*

--40 year-old female

**Support to return to health.** Some of the return to physical health was facilitated by the provision of ancillary services—many of which are not usually included as part of routine RR-TB care. As one participant reported when she described her difficulty walking and how she shared her worries about this with a nurse:

*"She said I shouldn't be worried, they'll organize physio for me so that I would be able to walk again. Staff from physio department came and they helped me exercise, stretching and they also gave me something to help me walk because I could not walk. Then I began to try to walk with crutches, and I could walk with those crutches to the bathroom to bathe. I could wake up during the night and take the crutches and go to the bathroom to pee, and I was no longer on a diaper."*

--30 year-old female

Some participants reported they were able to continue with the difficult therapy because of the social support they received during therapy. As one participant noted, when talking about the difficulties he faced during treatment:

*"It did cross my mind a lot, 'that it is beyond my powers, I am quitting' and even at home I was telling them that I was quitting. My wife motivated me saying, no, 'this is your life' and it also came to my mind that she was right, because of how sick I was. Yeah, I got motivation from home to finish the treatment. That is where I got enthusiasm, but if I was here, or I was not going to go back home, I was going to go somewhere else or disappear because it was tough."*

--62 year-old male

For others, it was the hope of re-establishing strong family or social connections after they were "cured" of their RR-TB that enabled them to continue with their treatment. As one participant noted:

*"When I was still in the ward here having a conversation with another patient asking her, 'what would happen to me if I can stop taking these pills? She said, 'oh no! you already started! Don't stop them, your children!'I said to her, 'no, I feel like they are harassing me.' She said, 'no, you mustn't stop them, this is your life, think about your children'. Then I continued to take them."*

--47 year-old female

Others reported that the medical staff at the treating facility were a major source of support and, in fact, the nurses were their most informed "guides" leading them through this confusing and difficult time period:

*"What I can say is that nurses cared for us here, and my wife cared a lot for me when I was at home. Yes, she was always next to me and pleading with me saying, 'you choose to leave your*

*children instead of trying these pills, it's better to say that you tried and failed instead of saying that you stopped them'."*

--52 year-old male

Another participant stated:

*"When I came here [treating hospital], nurses would crush and feed me pills, and they fed me food as well because they were caring. Today I can speak and I'm alive but I had lost hope of recovering, the way things were happening with me. [Nurses] from here are very caring. I've never been in a hospital where there are caring sisters like here. Even during the night when they noticed that I was restless I couldn't sleep, maybe I was feeling pains, they would come and check up on me. . .they took good care of me in this hospital, that's why I survived and I'm alive today"*

-- 30 year-old female

Following the advice and instructions of the nurses was perceived as the clearest path back to health by many participants:

*"I had that hope that if I do what the nurses tells us to do, I will be fine, because they would encourage us and say we will be alright. If we follow what they tell us to do we will be fine."*

--35 year-old female

Religious faith and spirituality were reported by some participants as a major source of support throughout their treatment journey:

*"I just said I pray that God will help me finish this treatment, until they stop it. Although it was painful and sad, but I did not lose hope. I had that hope that if God doesn't want to take you, it means your time has not come, but when that time comes I will just leave."*

-- 70 year-old female

One participant expressed his relief at completing his treatment in the following way:

*"When I came out of this I just said, 'Amen oh Lord, thank you.'"*

--52 year-old male

Finally, several of the participants talked directly about their suffering with RR-TB and the side effects of treatment knowing it was transient, and it was this "impermanent" state that made it bearable. As one participant stated:

*"It was not easy to take it, it's too much and it's hurtful. You wished you could stop it but then you tell yourself that because it's for a period of time and not for life-long, then I'll persevere."*

--30 year-old female

**Death a constant companion.** For some, it was the fear of death that motivated them to continue taking treatment. This was reinforced by hearing about or witnessing the deaths of other patients, events which were experienced as traumatic:

*"You see, I was distraught when I saw a patient that was opposite my bed die. I had to ask for a pass-out because I was so scared after I saw her dying in front of me. It was at night when I came back from the bathroom when that happened. I couldn't sleep after that incident. I sat on a chair until the morning. The nurses came at 4 o'clock to bath those that couldn't bath themselves. The sisters asked me and I said, 'no sister, could you please ask the doctor to grant me a pass-out to go home for at least one day.' I told them that I saw a person die here and I am terrified. The doctor granted me pass-out and I went home, and I came back the following day not as terrified as I was the previous day. But my heart had not forgotten what I saw happen in front of me."*

--30 year-old female

Another participant reported:

*"I never had that bad luck where somebody die in my ward, but in other wards there were* [and] *you think that you are next. . .and think that maybe tomorrow it will be me. . . Eish, my sister what I know is that I was fighting for my life, because most of the time we were told that if we stop treatment, nurses don't have a problem but most of the people died"*

--33 year-old male

Death was a constant threat for people who were diagnosed with RR-TB, and many of them described their illness as a suspended state between life and death, in which the treatment was the path they had to walk down to return to health. Although for many this was a psychological construct, others described it as a physical and social reality:

*"Yes, most of them feared me, and the things they were saying, I heard they were saying eish, 'it's just a matter of time'. That, 'it's just a matter of time, he's leaving, he's dying'. . . I remember another day my children went to buy candles in the shop. They use it to make floor polish. People who saw my children buying candles spread rumors that I have died, as my children were seen buying candles. And people who came to my house, they came to express their condolences."*

--52 year-old male

Several participants reported that it was only their care and concern for their children that enabled them to keep taking the treatment:

*"If I quit the treatment, it will be death and she* [my daughter] *will be left to suffer."*

--33 year-old male

## Well-being restored? Post-TB treatment health state

Participants all reported physical, social, psychological and economic changes as consequences of their RR-TB disease and treatment.

**Physical changes.** Many challenges reported by participants were physical, including a diminished ability to perform activities they were able to do prior to their RR-TB disease and treatment:

*"I am the kind of person who likes cleaning the yard and the flowers. You see, I have flowers in my room, palm trees, I make all those things. So, I no longer do all those things, I don't have the strength."*

--33 year-old male

Another stated:

*"My heart longs to do something but my body doesn't comply, you know there is just that feeling that my body got tortured somewhere"*

--35 year-old female

Another felt so physically different that she reported:

*"Because even my body, this is not my body. . ..I have lost weight, my clothes don't fit. . ..I am well, but it is not the same as before."*

--47 year-old female

For other participants, the side effects of the treatment caused permanent damage, as noted by one participant, who received an injectable agent known to cause deafness:

*"Yes, they saw that the eardrum has been destroyed, so it cannot draw sounds well. It's still painful even now. . .sometimes I struggle to hear it I turn and look on this side. They are saying that my hearing was severely damaged. It happened when I was here, it's the injection."*

--43 year-old female

Some participants, however, reported that their physical health returned to how it was before their RR-TB diagnosis:

*"I can say that there is no change because now I am very well like I had been before I got sick."*

--35 year-old female

Another participant noted that all was well once she completed the treatment:

*"What was difficult is that, taking pills was really hard, they make you tired, they make you choose what to eat, they make you moody, but when I finished treatment I went back to my normal self."*

--30 year-old female

For participants with long-term physical problems, a consequence of their RR-TB disease and treatment, there was no clear place to turn for ongoing support with managing these health issues. As one participant stated:

*"Basically, that doctor has done her job because in my understanding she deals with MDR, she's done. Doctors don't heal dyspnoea, not the doctors. I have been to doctors who are paid but they would say they don't know. Some doctors give you pills saying they not even sure if it (dyspnoea) will stop."*

--30 year-old male

**Social, economic and psychological changes.** Some participants also noted social, economic and psychological disruptions to their lives after successfully completing their TB treatment. One participant described how long-term physical changes affected his social life post his RR-TB disease and treatment:

*"I go to Zion church, we go around the circle while singing, those are the things I could not do anymore. I like doing it, when I try to go around the circle and sing the dyspnoea would stop me then I will sit down. I would feel sad that I can no longer do what other man do. I would feel sad."*

*"Its going out I miss. You know that as men we like meeting together and take a walk in the township as I am from the township. Just to go out and wander around. I can't do it and spend a lot of time alone."*

--33 year-old male

Another participant reported that both her brother and his family and her partner left her because of her RR-TB diagnosis and treatment experience:

*"As I am sitting outside, people passing by would laugh and laugh, but then I didn't have a problem with that because I knew what I wanted. And those around me it was really hard. Because we were staying at home with my brother and his wife and his children. So when I came back from hospital they moved out and they said they bought their own house. They left like that, and it was my very first days after I was discharged and came back home. Then at home I was left with my child the two of us. And my partner left me."*

--35 year-old female

Another participant reported that there were changes in the way others acted towards them:

*"When I was discharged from here I was very well and could feel it. . .But it has affected my life, because you notice that people don't like you as before because they think that you will infect them. I explained to them* [clearing throat] *and they understood because I told them 'now that I am discharged, you don't get discharged if you are not well', which means that I am well because I can stay with people at home. . ..but they are not sure because I still go for check-ups. Some understood, but some did not take it very well, you find that they would say, 'no, what he's telling us is not true.'"*

-- 33 year-old male

One participant reported long-lasting issues with sexuality that began during treatment, stating:

*"There was a decrease in sexual desire. . .you know, when there is change that happens in the bedroom, they tend to think that there is something suspicious."*

--62 year-old male

Another reported transient issues with sexual relationships because she feared that an intimate relationship with a man was how she first got infected with RR-TB:

*"Something that I can say happened, I did not want a man near me. I can say that maybe it was happening because I had these thoughts in my mind that I don't know how I got infected with this TB. I had those thoughts that maybe I got infected because of him. I had those thoughts, that's why I did not want him near me."*

--35 year-old female

In contrast, some participants reported their relationships were maintained or even strengthened, especially if the others in the family had their health assessed and were found not to have RR-TB. When asked if her relationship with her husband had changed one participant responded:

*"No, it did not, because we are still in love even now, and he went for check-up, but he did not have TB."*

--43 year-old female

However, even for those who were able to maintain their social relationships—and many participants reported receiving high-quality support from those in their networks—there was still an experience of a social loss:

*"People who were my friends in my neighbourhood, when I got home, they came to support me. Because when I got home, I would tell them that I am unable to go to them and ask them to come see me. When I was getting the 3 days pass-outs they would come over with juices or anything that I could eat, they were with me. Together with the church members and my family. So, they never distanced themselves from me, they did not insult me. It was just me who felt sad about being unable to do anything that I like. . . . . .that's how I felt."*

--33 year-old male

Participants also reported losing their livelihoods as noted by one woman who developed painful peripheral neuropathy in her feet as a side effect of her treatment:

*"Oh! I stopped selling long ago, because of my feet. My boy would carefully help me and bring me here because he has a car. I was nothing and helpless at that time. I was using this to walk, my walking stick. I am stronger now, but never strong enough to sit all day and sell, eish."*

--70 year-old female

Another participant reported:

*"My work was forced to be destroyed, my work had to stop and totally shutdown. I could not even handle or carry a spade. If I carry it I would be shaking, even a spoon, I would be shaking. . . . . . . . .Sometimes I would be forced to go and work in some households that needed help at that time, and when the person saw me like that, he would feel sorry for me and say, 'no, man, no, take this R20, just go home and don't work here because you will die here in the field, then I will be arrested'*

--50 year-old male

One participant also reported that she was no longer able to pursue her education because of her RR-TB diagnosis and treatment, noting:

*"I was studying, yes, I was studying.* [Treatment] *destroyed me because I was left behind. And it happened that the person who was teaching me died. My studying died too."*

--43 year-old female

Some participants reported positive mental health and aspirational changes after being successfully treated for RR-TB, but even these were mitigated by long-lasting deleterious consequences of treatment:

*"The change in my life, sister, is that I wish to help the community by teaching them about being sick, and about loving and caring for yourself. I have seen hardship. I want to do nursing so that I can help young people like me, and talk about something that I know, that I've been through. What stopped me was my ears, I have impaired hearing as a result I am staying at home."*

--30 year-old female

Despite these long-lasting consequences, some of the participants reported that they were happy with their post-treatment states, relieved that the disease and arduous treatment were over:

*"No, I would say it is slowly becoming better now, because lots of things that I wasn't able to do before (when I was sick) I can do them now and things are going back to normal. . .. it is slowly approaching 100, maybe-70 [out of 100] or something. . .. slowly going back to normal. Even my complexion is improving, I was extremely dark like this phone."*

--35 year-old female

Another participant reported:

*"But when time went by I realized that I was better because I could walk, because I could walk to town by myself. That's when I realized that there is a difference."*

--35 year-old female

**Spectre of TB returning.** Despite returning to relative health and moving on in their lives, the experience of having RR-TB was so traumatic that participants worried they could have another episode of RR-TB. This caused them considerable anxiety and fear:

*'Eish it happens* [thinking of getting RR-TB again], *it happens a lot. As a result I have decided that if it comes back, I'd rather kill myself. It brought so much pain into my life I always say if it comes back, I'd rather die. I choose death over it. It's hard, it's really hard.'*

--52 year-old male

*'I was terrified when I stopped taking the treatment, as I didn't want to get so sick again. I wanted to keep taking the pills forever.'*

--33 year-old male

*'It worries me a lot that the TB might come back and stay here. Oh! God forbid! 'If I see a person who doesn't look too well, I look at him and say "Ohhh no, this one might just take me back to where I was."*

--35 year-old female

## Discussion

For the people who participated in this study, surviving RR-TB was a transformative experience. The development of RR-TB disease and its treatment were major life disruptions and the experience had multiple long-term consequences. The disease and the journey back to health

had a lasting impact on the physical, economic, social, and psychological aspects of people's lives. For some, these long-term consequences were positive, but this was not the case for all participants. In the current health system, the primary focus of RR-TB treatment is bacteriologic outcomes and individuals are dismissed from care services once these biological endpoints are met, regardless of their ongoing health or social needs.

For all participants interviewed, coping with an episode of RR-TB disease required considerable inner strength and determination. However, even more strength and inner resolve were needed to endure the lengthy toxic treatment and initial period of hospitalization. Many of our study participants had an injectable agent included in their 24-month long treatment regimen and were hospitalized for some time. In recent years, in many countries, the injectable agent has been replaced with an all oral regimen, the length of treatment and hospitalisation have been reduced and decentralised models of care have been introduced [30]. However, the disease and treatment challenges are still multifaceted and difficult to endure, as people with RR-TB are often from the most socio-economically deprived and vulnerable households [31, 32], many are unable to stay the course of treatment. Addressing the social determinants of health is key to TB control [33]. During the COVID-19 pandemic resources were mobilized to provide nutritional and economic support [34].Mechanisms to deliver this aid were established and should be explored for the continued delivery of support for people with RR-TB. The COVID-19 pandemic has also led to the creation and development of several virtual supportive adherence counselling and clinical management systems including links to health services via toll-free helplines, links to practitioners using virtual systems and the creation of virtual WhatsApp support groups. If integrated into the existing digital and mobile health technologies (mHealth solutions), effective remote counselling and psychological support could be a component of the support offered to RR-TB survivor networks.

Several study participants reported the value of support from other patients with RR-TB. This suggests that support groups convened by RR-TB survivors, together with professionals e.g. psychologists or social workers could play a role in the management of RR-TB following treatment completion. Such groups could provide an opportunity to discuss the complexities of re-integration into a community after an episode of RR-TB, the fear of getting TB again and the fear of death, Group discussions have been effectively used in low and middle income countries to enable participants to discuss and share their experiences of stigma, discrimination and pain, where they have experienced support and some relief [35, 36].

Participants also reported that nurses were a major source of support and guidance for them during their liminal period as they were being treated for RR-TB. Nurses have long been recognized as key providers in the management of RR-TB [37, 38]. Their crucial work, however, is often not formalized nor recognized by the larger health system as a whole, which can result in burn-out and sub-optimal service delivery [39]. This study suggests that nurses would be a natural cadre to lead programs for the long-term care of people who have been treated for RR-TB, provided they are trained, supported and recognised for doing so. They could be supported by RR-TB survivors employed as peer supporters or health educators.

After successful treatment completion the World Health Organization (WHO) and South African RR-TB guidelines recommend follow-up for one and two years respectively to ensure recurrent RR-TB is detected early and appropriate treatment started where indicated [40, 41]. In reality, these visits rarely happen for a variety of reasons, many related to an over-burdened health system which does not prioritize people who have been "cured". With renewed commitment and investment, however, these follow-up appointments could include counselling and strategies to address post-treatment care for physical ailments, social re-integration and how the perceived fear and risk of getting TB again could be mitigated. Given how common physical disability was in the people who participated in this study, lifelong medical support and

rehabilitation may be necessary and should be considered a part of routine TB care, provided to "successfully treated" patients free of charge [42].

There are multiple limitations to this study. Chief among them is recall bias, since several participants were several years removed from their actual treatment experiences. They may have remembered, and conveyed events differently compared to when those events occurred, and this could have had an impact on our study results. A second limitation is the small sample size from a single institution whose experiences are unlikely to be generalizable to other populations. A third limitation is that of the original 27 individuals identified as potential participants, only 15 were able to be contacted and 12 agreed to participate in the study meaning there is likely a sampling bias toward those individuals who could be reached by the research team. It could be that individuals who were not able to be contacted were more fully integrated back into society and may have had fewer post-treatment complications. Additional work is clearly needed with other populations, and it is hoped our study findings can help guide larger studies on this important topic. Finally, as part of the important tradition of reflexivity in qualitative research, we note that two of us are involved in providing treatment services to people living with RR-TB, and this may have biased our interpretation of the study data.

## Conclusion

Despite these limitations, this qualitative research shows that the diagnosis and treatment of RR-TB is a transformative experience for people who have survived the disease. Once a diagnosis is made, those with RR-TB enter a liminal period in which their lives are marked by great uncertainty characterized by physical and mental suffering. Their journeys back to health can be treacherous but with the support of their social networks, the nursing staff, and other patients, they can return to some degree of normality. Most of them, however, have long-term physical, social, economic, and psychological consequences which are not addressed by short-sighted TB programs that are built around bacteriologic outcomes that narrowly define treatment success. A more holistic and long-term view of post-TB health—including the provision of comprehensive medical and social services for a range on ongoing issues—should be a central part of person-centred TB care.

## Supporting information

**S1 Checklist.**
(PDF)

**S1 File.**
(DOCX)

## Acknowledgments

The authors of this paper are incredibly thankful to the men and women who participated in the interviews. We are thankful to Dr. Iqbal Master for inspiring this project and so much of the work we all do in the field of tuberculosis.

## Author Contributions

**Conceptualization:** Marian Loveday, Lee-Megan Larkan, Helen Cox, Johnny Daniels, Erika Mohr-Holland.

**Data curation:** Marian Loveday, Sindisiwe Hlangu, Johnny Daniels.

**Formal analysis:** Marian Loveday, Sindisiwe Hlangu, Lee-Megan Larkan, Helen Cox, Johnny Daniels, Erika Mohr-Holland, Jennifer Furin.

**Methodology:** Marian Loveday.

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
