## [Decision Letter · Decision Letter 0]

22 Mar 2021

PONE-D-21-02389

"This is not my body:" Therapeutic Experiences and Post-Treatment health of people with rifampicin-resistant tuberculosis

PLOS ONE

Dear Dr. Furin,

Thank you for submitting your manuscript to PLOS ONE. After careful consideration, we feel that it has merit but does not fully meet PLOS ONE’s publication criteria as it currently stands. Therefore, we invite you to submit a revised version of the manuscript that addresses the points raised during the review process.

Sorry that it has taken this long to get a decision to you. I found it difficult to get reviewers for this paper - several people declined because they have collaborated closely with one or more of the authors. 

I would ask you to address the reviewer’s comments, particularly the request for more details of the sampling strategy, recruitment process etc. In addition to the reviewer’s comments I have a couple of other comments that I would ask you to consider:

I couldn’t see the SRQR or COREQ checklist for reporting of qualitative research. Please could you ensure that reporting is in line with one or other, and that a completed checklist is submitted with the revised manuscript.Please explain clearly in the manuscript when the interviews took placeIt seems that all participants were hospitalised initially for treatment. When addressing the reviewer’s comment about sampling strategy, please clarify whether or not this was a specific inclusion criterion. It seems that recruitment took place over a period when some people with RR-TB/MDR-TB would have been treated on an outpatient basis from the start of treatment, so if you intentionally only recruited people that were hospitalised this should be explicit

We look forward to receiving your revised manuscript.

Kind regards,

Richard John Lessells, BSc, MBChB, MRCP, DTM&H, DipHIVMed, PhD

Academic Editor

PLOS ONE

Journal Requirements:

2. When reporting the results of qualitative research, we suggest consulting the COREQ guidelines: http://intqhc.oxfordjournals.org/content/19/6/349. In this case, please consider including more information on the number of interviewers, their training and characteristics; and please provide the interview guide used.

Reviewers' comments:

Reviewer's Responses to Questions

**Comments to the Author**

1. Is the manuscript technically sound, and do the data support the conclusions?

Reviewer #1: Yes

2. Has the statistical analysis been performed appropriately and rigorously? 

Reviewer #1: N/A

3. Have the authors made all data underlying the findings in their manuscript fully available?

Reviewer #1: Yes

4. Is the manuscript presented in an intelligible fashion and written in standard English?

Reviewer #1: Yes

5. Review Comments to the Author

Reviewer #1: Thank you so much for the opportunity to review this manuscript. The title is very interesting and the study researched an important area in patient health - post treatment health of patients. I wonder if the authors would consider focusing a large part of the paper on post health experiences, as opposed to including treatment experiences. I realise that this would be a huge change in the contents of the paper and would not object to leaving it as is.

Abstract

The results section of the abstract is rather long, perhaps some information could be left out (line 65 - 68).

Data collection and analysis

Line 150 refers to Appendix 1, I could not find this attachment.

The sampling method used is rather unclear. Sampling procedures were not clear, for instance how the researchers ended with the 12 participants who were found in the national register, how contact was made, who approached them, where the data was collected, at the study site or at home?

in the presentation of the data, under "well-being disrupted", I noticed that the authors present quotations from male participants, and in the next section "returning to well-being", female voices are presented. Was this deliberate, or a coincidence? perhaps both gender may be represented in both sections.

Line 236 the sub-heading "side effects" could be rephrased, it seems the issues are physical issues, perhaps a word that may reflect the quotes better.

The sentence on line 380 that patients saw others die due to not taking their treatment seems to be the opinion of the authors and not what the patients reported. The cause of death may be from reasons other than not taking their medication, perhaps the authors can rephrase this.

Line 415 consider removing the subtitle "additional motivations"

Line 471-479. The quotation does not adequately support the assertion that patients did not know where to turn for ongoing support. It seems that patients did know where to go but preferred to be treated at the facility where they had been treated for TB. It also shows the level of trust that they had on their healthcare workers at Greytown Hospital. Perhaps the authors can use the second quotation as support for this assertion.

Line 627 please add "I" in the quotation between ...back to where was

Discussion

Line 662. This sentence is rather long and lost some of its meaning, consider rephrasing.

The conclusion and recommendations are very good and feasible.

Thank you so much

6. PLOS authors have the option to publish the peer review history of their article (what does this mean?). If published, this will include your full peer review and any attached files.

Reviewer #1: **Yes: **Dr Boitumelo Seepamore

---

## [Author Response · Author response to Decision Letter 0]

6 Apr 2021

Comment Reply

I couldn’t see the SRQR or COREQ checklist for reporting of qualitative research. Please could you ensure that reporting is in line with one or other, and that a completed checklist is submitted with the revised manuscript.

 We have added in the COREQ guideline. Forgive us for this oversight. 

Please explain clearly in the manuscript when the interviews took place

 We added that the interviews took place in a private office setting. 

It seems that all participants were hospitalised initially for treatment. When addressing the reviewer’s comment about sampling strategy, please clarify whether or not this was a specific inclusion criterion. It seems that recruitment took place over a period when some people with RR-TB/MDR-TB would have been treated on an outpatient basis from the start of treatment, so if you intentionally only recruited people that were hospitalised this should be explicit

 Thank you for this comment. Study participants were recruited from patients treated for RR-TB from 2008 – 2018. During this time, although decentralized care for RR-TB was available in many settings in South Africa, in this study setting, all patients were initially hospitalized, to initiate treatment, before being discharged to the health centers. We have added this information into the manuscript. 

If there are ethical or legal restrictions on sharing a de-identified data set, please explain them in detail (e.g., data contain potentially identifying or sensitive patient information) and who has imposed them (e.g., an ethics committee). Please also provide contact information for a data access committee, ethics committee, or other institutional body to which data requests may be sent. These are qualitative interviews and they do contain information that could be identifying. Thus we cannot publish them or make the widely available since this could lead to inadvertent recognition of the participants, since the rich quotes and descriptions cannot be fully anyonymized. Our ethics committee is willing to share the data with individuals who request it. Contact can be made by reaching out to Ms Anitha Gupta (email: Anitha.gupta@mrc.ac.za) at the ethics board at the South African Medical Research Council

I wonder if the authors would consider focusing a large part of the paper on post health experiences, as opposed to including treatment experiences. I realise that this would be a huge change in the contents of the paper and would not object to leaving it as is. We appreciate this comment from the reviewer. We ourselves were surprised about how often the participants wanted to discuss their treatment experiences, and it was our interpretation that these experiences framed their post-treatment experiences. Thus we felt it was important to present these treatment experiences as well since it would have been difficult to understand the post-treatment experiences without them. We have added a clarifying comment on this which states: “Of note that while the interviews were focused on the post-treatment experiences of the participants, most participants framed such experiences within the context of what happened to them during treatment. Thus we also present information on treatment experiences as a way to frame the post-treatment experiences of the participants.”

The results section of the abstract is rather long, perhaps some information could be left out (line 65 - 68). We have removed lines 65-68.

Line 150 refers to Appendix 1, I could not find this attachment. We have now included this—our apologies for the omission. 

The sampling method used is rather unclear. Sampling procedures were not clear, for instance how the researchers ended with the 12 participants who were found in the national register, how contact was made, who approached them, where the data was collected, at the study site or at home? In the section on study design (lines 138-139) we describe how individuals were purposively selected to ensure both males and females were represented and that there was representation across the study period. To further clarify the sampling method, we have included in lines 151-154 how many people were initially contacted, how many declined to be interviewed together with their reasons for declining. 

In the presentation of the data, under "well-being disrupted", I noticed that the authors present quotations from male participants, and in the next section "returning to well-being", female voices are presented. Was this deliberate, or a coincidence? perhaps both gender may be represented in both sections. This was not done deliberately but was by chance. We appreciate this being pointed out and have changed some of the quotes provided. 

Line 236 the sub-heading "side effects" could be rephrased, it seems the issues are physical issues, perhaps a word that may reflect the quotes better. We have change this so it now says “Physical Issues”

The sentence on line 380 that patients saw others die due to not taking their treatment seems to be the opinion of the authors and not what the patients reported. The cause of death may be from reasons other than not taking their medication, perhaps the authors can rephrase this. We have removed the words “who did not take their treatment” and the sentence now reads that “This was reinforced by hearing about or witnessing the deaths of other patients, events which were experienced as traumatic.”

Line 415 consider removing the subtitle "additional motivations" We have removed this subtitle.

Line 471-479. The quotation does not adequately support the assertion that patients did not know where to turn for ongoing support. It seems that patients did know where to go but preferred to be treated at the facility where they had been treated for TB. It also shows the level of trust that they had on their healthcare workers at Greytown Hospital. Perhaps the authors can use the second quotation as support for this assertion. We have removed the first quote from the manuscript.

Line 627 please add "I" in the quotation between ...back to where was We have added this back in. 

Line 662. This sentence is rather long and lost some of its meaning, consider rephrasing. We have edited this into two sentences and removed the question form so the paragraph now reads: “This suggests that support groups convened by RR-TB survivors, together with professionals eg. psychologists or social workers could play a role in the management of RR-TB following treatment completion. Such groups could provide an opportunity to discuss the complexities of re-integration into a community after an episode of RR-TB, the fear of getting TB again and the fear of death,”

---

## [Editor Report · Decision Letter 1]

28 Apr 2021

"This is not my body:" Therapeutic Experiences and Post-Treatment health of people with rifampicin-resistant tuberculosis

PONE-D-21-02389R1

Dear Dr. Furin,

We’re pleased to inform you that your manuscript has been judged scientifically suitable for publication and will be formally accepted for publication once it meets all outstanding technical requirements.

Kind regards,

Richard John Lessells, BSc, MBChB, MRCP, DTM&H, DipHIVMed, PhD

Academic Editor

PLOS ONE
---

## [Editor Report · Acceptance letter]

30 Apr 2021

PONE-D-21-02389R1 

*“This is not my body”:* Therapeutic Experiences and Post-Treatment Health of People with Rifampicin-Resistant Tuberculosis 

Dear Dr. Furin:

I'm pleased to inform you that your manuscript has been deemed suitable for publication in PLOS ONE. Congratulations! Your manuscript is now with our production department. 

Kind regards, 

on behalf of

Dr. Richard John Lessells 

Academic Editor

PLOS ONE